# Prevalence, risk factors and health seeking behaviour of pulmonary tuberculosis in four tribal dominated districts of Odisha: Comparison with studies in other regions of India

**Tahziba Hussain**[1]*, **Sushri Shanta Tripathy**[1], **Shritam Das**[1], **Prakasini Satapathy**[1], **Dasarathi Das**[1], **Beena Thomas**[2], **Sanghamitra Pati**[1]

**1** ICMR-Regional Medical Research Centre, Bhubaneswar, Odisha, India, **2** ICMR-National Institute for Research in Tuberculosis, Chennai, India

* tahziba_hussain@hotmail.com

**Data Availability Statement:** All relevant data are within the paper and its Supporting Information files.

## Abstract

### Aim

To determine the prevalence of pulmonary tuberculosis, socio-cultural practices and health seeking behaviour of tribal people in four districts of Odisha.

### Methodology

This was an action research study with qualitative and quantitative design following a sequential approach implemented in a 4-phased manner. It was carried out in the 6 selected villages from July,2015 to June,2017. The screening for active TB among chest symptomatics is followed as per the guidelines of the (RNTCP) Revised National Tuberculosis Control Program in India.

### Results

In all, 1455 households were surveyed in the 6 tribal dominated villages of 4 districts, namely Balangir, Dhenkanal, Kandhamal and Mayurbhanj. Total population of the villages was 6681. Based on the eligibility, 5144 (97.7%) individuals were screened. About 139 (2.3%) could not be screened due to non-availability in their households during day time. Out of the screened individuals (5144), 126 chest symptomatics were identified. Sputum samples were collected from them and sent to the National Reference Laboratory, RMRC, Bhubaneswar using public transport and maintaining cold chain. Out of 126 chest symptomatics, 35 patients were found to be having active TB disease and 18 were culture positive. The prevalence of pulmonary TB is 0.68%. The risk factors seemed to be ignorance about TB symptoms, addiction to alcoholic drinks, difficulty reaching the health facilities owing to the long distances, lack of communication and transport. In addition, other morbidities like

**Funding:** ICMR Tribal Task Force funded this project grant (Tribal/89/TB-15/2014-ECD-II) to Dr. Tahziba Hussain of Regional Medical Research Centre, Bhubaneswar. The funder had no role in study design, data collection and analysis, decision to publish or preparation of the manuscript.

**Competing interests:** The authors have declared that no competing interests exist.

Malaria, diabetes, hypertension, malnutrition, etc. were observed in the tribes of the study sites.

## Conclusion

TB control programs need further strengthening in the tribal dominated regions. This study is the first of its kind in this State.

## Introduction

In India, the tribal population, is numerically a small minority but represented by an enormous diversity of groups. Their language, linguistic traits, size of the population, physical features, acculturation, modes of livelihood, social stratification and stage of development varies among themselves. Tribal communities are spread across the country but live in different ecological, geographical and climatic conditions including plains, forests, hills and inaccessible areas. Majority of the Scheduled Tribe population is concentrated in the eastern, central and western region including the nine States namely, Andhra Pradesh, Chhattisgarh, Gujarat, Jharkhand, Madhya Pradesh, Maharashtra, Odisha, Rajasthan and West Bengal. Although tribal groups have adapted to the mainstream of life but still are at different stages of socio-economic and educational development [1, 2, 3, 4].

In India, the state of Odisha has 62 distinct tribal groups, having the highest number of tribes, largest collection of tribal people consisting of 24 percent of the total population of the State. Each of these tribal groups has its own indigenous customs and continues to practice them even today. Majority of tribal groups are dependent on agriculture as labourers or cultivators. Some of the Scheduled Tribes do not follow their traditional occupations and work as migrant labourers in mines and factories. Subsistence oriented economy of the tribes revolves around forests and is based on food gathering, hunting and fishing. Even the large tribes like Gond, Oram, Munda and Santal have settled as agriculturists but still supplement their livelihood with hunting and gathering. Majority of their population is concentrated in four districts of Balangir, Dhenkanal, Kandhamal and Mayurbhanj [5, 6, 7]. India has the highest burden of tuberculosis (TB) with 23% of the global burden of annual incidence of active TB patients. The estimated incidence of TB in India is about 28,00,000 accounting for about a quarter of the world's TB cases as per the Global TB report 2017 [8].

TB incidence in Odisha in 2017 was estimated at 159/lakh/year vis-a-vis national average incidence of 138.33/lakh/year. Odisha figured among top-ten TB incidence States in the country. District-wise details reveal that Gajapati has the highest incidence of 275/lakh/year in State and is followed by Mayurbhanj, Malkangiri, Rayagada and Sundergarh. Annual Total Notification Rate of TB in the districts of Balangir, Kandhamal, Mayurbhanj and Dhenkanal is 118, 203, 268 and 147, respectively, [9, 10]. There are hardly any epidemiological studies on the prevalence of TB among the tribes in Odisha.

Therefore, this study was conducted to determine the prevalence of TB by active case finding among the tribes living in the six tribal dominated villages of Balangir, Dhenkanal, Kandhamal and Mayurbhanj. This study aimed to improve TB case detection, treatment compliance and provide quality TB care through a community based approach by involving the villagers, health care workers and organizing FGDs and awareness meetings. The socio-demographic profile, risk factors and health seeking behaviour were assessed by situational analysis, house-hold survey and individual interviews. Individuals were screened and chest

symptomatics were identified. Sputum samples were collected from them and processed for detection of *M.tuberculosis* in the National Reference Laboratory (NRL) at Bhubaneswar. TB positive cases were referred for treatment at the nearest DOTS centre. The functioning of Revised National Tuberculosis Control Program (RNTCP) in Designated Microscopy Clinic (DMC), Tuberculosis Unit (TU) and District TB Centre (DTC) in tribal areas of Odisha were reviewed to identify gaps in program implementation. For this, interviews of Medical officers, STS, ASHAs and other health workers of the selected villages of the districts were conducted.

This study is the first of its kind in this region of the country.

## Materials and methods

### Ethical approval

State TB Cell, Directorate of Health Services, Odisha at Bhubaneswar provided permission to conduct the study in the selected districts. The study was approved by the Institutional Human Ethics Committee. Written consent was obtained from each individual. The questionnaires were translated in local language, Odiya. TB cases detected during the survey were counseled and referred to the nearest RNTCP facility for initiating Anti-Tubercular Treatment (ATT). Follow up of the TB patients was done during repeat visits to the villages in the intervention phase. Individuals not having TB but with symptoms were referred to the local health center.

### Study period

This study was carried out during Oct.,2015-June,2017 in the six tribal dominated villages of Balangir, Dhenkanal, Kandhamal and Mayurbhanj. The villages selected for the study are Maghamara under Patnagarh Police Station in Balangir district. Jantaribola under Kamakhya-nagar Police station in Dhenkanal district. Penagaberi under Tikabali Police Station in Kandhamal district. Kasiabeda & Bhadua under Jharpokharia Police Station in Mayurbhanj district. Gandirabeda under Jashipur Police Station in Mayurbhanj district. Prior to the survey, the field investigators visited the selected villages of each district to build a rapport with the health workers, village leaders and the tribal community to seek their support.

### Study investigators

Project Investigator supervised the overall survey. The team of trained field investigators and Technicians did the enumeration of residents in the households, screening by interview, sputum collection, examination by smear and culture and analysis.

Study participants were the tribals of Balangir, Dhenkanal, Kandhamal and Mayurbhanj.

### Sampling design and procedure for quantitative survey

Tribal villages with tribal population >70% formed the sampling frame for selection of villages. Villages within each zone were selected based on Probability proportional to size (PPS) method.

### Sample size

The required sample size was estimated to be about 2400 adults aged ≥15 years selected from 6 clusters for an assumed prevalence of 387/1,00,000 bacteriologically positive Pulmonary TB (PTB) cases, with a precision of 15% at 95% confidence level, a design effect of 1.3 and missing or non-response of 10%.

## Study design

This was an action research study with qualitative and quantitative design following a sequential approach implemented in a **4-phased manner** (Fig 1).

(i) The first phase was the **formative phase** during which **situational analysis** was done. This included an enumeration of the selected samples for the study from the secondary data available [geographical mapping, health facilities available, the distances between health

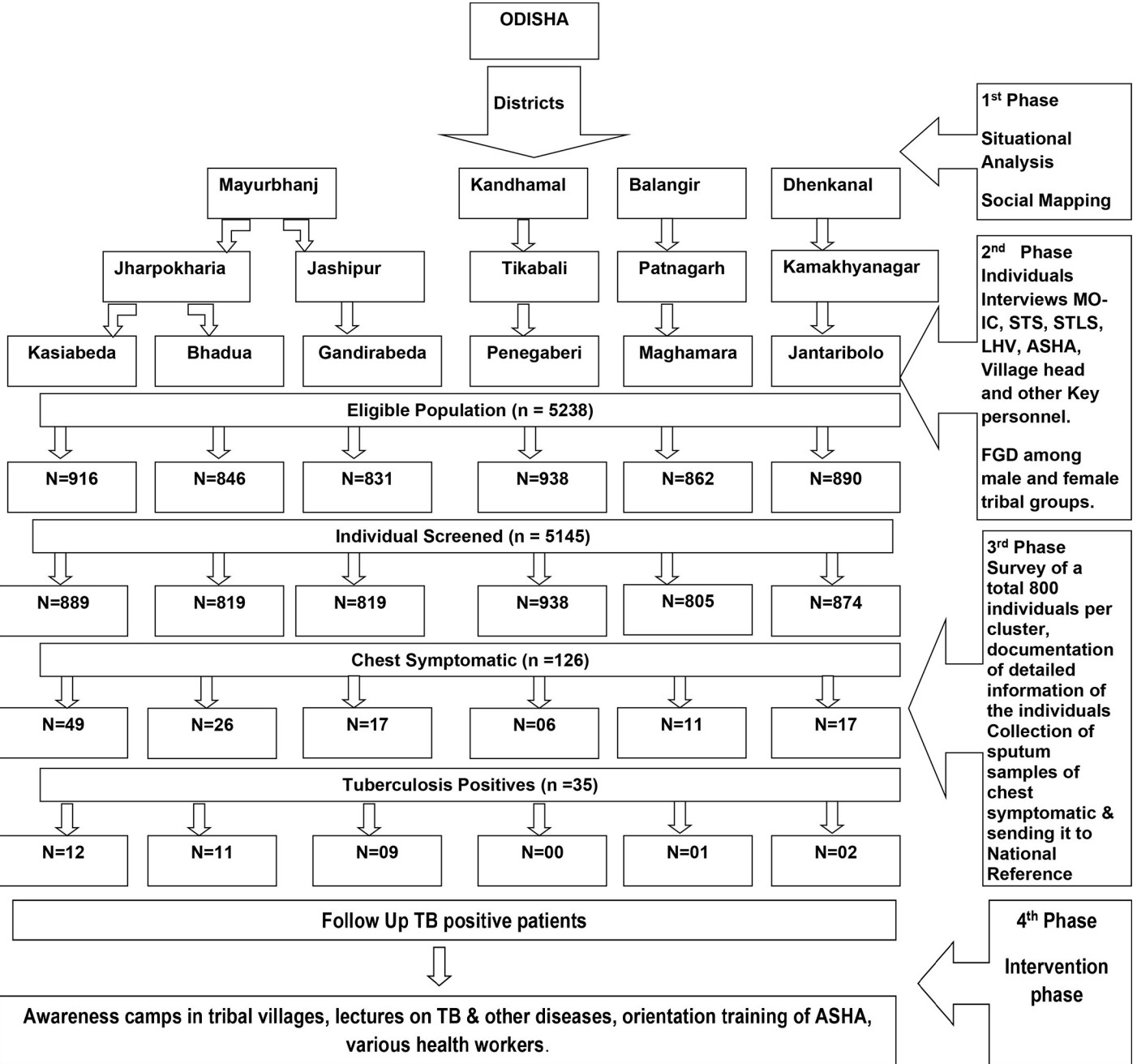

**Fig 1. The schematic presentation of the study carried out in phases.**

facilities available (private, public) and the socio-demographic profile of the tribal population [ethnicity and the number of households].

## Situational analysis

Field visit and social mapping of the six selected clusters namely Maghamara (Balangir), Jantaribola (Dhenkanal), Penagaberi (Kandhamal) and Bhadua, Gandirabeda and Kasiabeda (Mayurbhanj) districts of Odisha were conducted. District information of all the selected clusters were collected.

In each village, a sketch map of lanes and hamlets showing the number of houses was drawn, after inspection and discussions with villagers. Survey was started in each village after mutual agreement. Investigators recorded the age, gender and resident status of each individual of each household. Eligible persons (>15 years) were registered into an individual card. Majority of adults in the villages were engaged in agricultural occupations and some worked in other states.

(ii) The second phase was Qualitative assessment and qualitative data collected through Focused Group Discussion (FGDs) and interviews with influential representatives within the population, heads and the older members of the tribe, TB patients, families of TB patients and public health providers namely Senior Treatment Supervisor (STS), Senior TB Laboratory Supervisor (STLS), Laboratory Technician (LT), Multi-purpose Health worker (MHW), Lady Health Visitor (LHV), Accredited Social Health Activist (ASHA) and Aanganwadi workers. These FGDs and interviews elicited information on the health care seeking behaviour patterns of chest symptomatic in the population, their perceptions on TB, the risk factors that increase vulnerability to TB, cultural beliefs and practices, especially with regard to illness in general and TB specifically, access to health care and the barriers in accessing care and challenges in providing care on the treatment and management of TB among the tribal groups. The functioning of the Revised National TB Control Program (RNTCP) was reviewed in the public health facilities namely Designated Microscopy Centre (DMCs), Tuberculosis Units (TUs) and District TB Centre (DTC) of these areas to identify gaps in program implementation. The health facilities were visited, available records were reviewed and the staff interviewed with the help of standard formats available under the RNTCP. The facilities at DMC and TU providing services to the clusters were visited and evaluated. For assessing district level interventions, the DTC were visited in each selected district.

(iii) The third phase was Quantitative Assessment which involved interviews with the head of the house hold from the randomly selected families using a structured interview schedule. This schedule was designed in 3 parts. Efforts were made to meet all adult members of the household for the interview. This was done at a time convenient for the members finalized during the situational analysis. This required early morning and/or late evening visits depending on the occupational profile of the villagers.

## Interview schedule consisted of 3 parts

i.  **Household form.** The first part consisted of 12 questions and covered socio-demographic details of each of the members in every household, the type and size of the houses, cooking mode (indoor/outdoor), fuel used for cooking, ventilation for smoke, etc.

ii. **Individual form.** The second part consisted of 61 questions and covered anthropometric details, education, occupation, habits (alcohol / drugs), etc. of the members, general health complaints, and health seeking behaviour (problems accessing health facilities, any other disease like Malaria, diabetes, etc. knowledge, awareness of TB and contact history of TB.

iii. The third part was for chest symptomatics identified during the household survey, symptoms in the previous 3 months like weight loss, chest pain, fever, loss of appetite, haemoptysis, chest symptomatic in the household, and included information on visits to health centres, type of health centre and the experiences with regard to distance covered, economic implications, investigations, number of visits to the centre, delay, perception of the health providers with regard to accessibility, availability and attitude of staff.

iv. The fourth part of the schedule covered the patients already diagnosed with TB during the visit to the household in the tribal areas. This included details of TB (pulmonary/extra-pulmonary), duration of the period till TB diagnosis, place of diagnosis, either undergoing treatment under RNTCP or not, details of the health provider, etc.

## Sample collection

Each eligible individual was screened for presence of signs and symptoms suggestive of PTB (persistent cough for ≥2 weeks, fever or chest pain for ≥1 month, haemoptysis) and history of ATT. As a quality control measure, 10% of the eligible individuals were re-interviewed. Individuals having pulmonary symptoms or a positive history of ATT were eligible for sputum examination. Sputum specimens were collected in a pre-numbered sterilized sputum cup on the spot and a pre-numbered empty bottle was given for collection of the other sample next morning. Thus, two sputum samples [one each of spot and morning] were collected in sterile tubes from all the symptomatic individuals in the village. Sputum containing bottles marked with identification of each patient were packed and transported on the day of collection to the accredited NRL laboratory of the Institute by public transport maintaining cold chain as per RNTCP guidelines. From both the samples, a smear was taken and stained by ZN / FM method and examined for acid fast bacilli. The remaining sample was processed for culture by LJ method as per manufacturer's protocol. All positive tubes were subjected to AFB staining (ZN) for confirmation of growth. A PTB case was defined as an individual whose sputum is positive for acid fast bacilli by ZN microscopy and/or growth of *M.tuberculosis* by culture examination followed by DST (Drug Sensitivity Testing).

## Intervention phase

Based on the findings of 3 phases, interventions were incorporated in the study. This involved a health systems approach to address the gaps identified in effective delivery of RNTCP services. This included case referrals, networking with the RNTCP to ensure that symptomatics who were diagnosed with TB are enrolled in the Directly Observed Treatment Short-Course (DOTS), contact screening and TB prophylaxis were recommended when TB patients have contacts below 6 years of age. IEC activities were conducted to enlist possible community DOTS providers and sputum collection centres in the villages using the indicators to identify its impact. In this, 7 lectures were arranged in 12 villages. 14 awareness programs were conducted, covering 19 villages. About 185 (250) villagers attended these programs. About 30 Orientation training / meetings were organized for the health care providers namely DTO, STS, STLS, MO-IC, ASHA and LHVs. Transmission, prevention, risk factors, RDK, referral, treatment, care, adherence, follow up, contact screening, prophylaxis, co morbidities, etc. of TB and Malaria were discussed.

## Procedures

Details with regard to any member of the family having symptoms suggestive of TB during the last three months were obtained. Details on any individual in the family having been diagnosed

to have TB (contact history) during the previous one year or currently under treatment at the time of the interview were obtained. The team also recorded the difficulties encountered, if any, to reach the villages in terms of distances, accessibility, transportation and connectivity issues [telephone, internet/network].

### Data analysis and statistics

The data were collected using a structured questionnaire and entered in Epi Data version 3.1 (The Epi Data Association, Odense, Denmark). Data analysis was done using SPSS Version 21.0 (IBM Statistical Package for the Social Sciences) Statistics for Windows, (developed by IBM Corp, Armonk, New York). The results were presented in proportion and percentages.

## Results

In all, 1455 households were surveyed in the 6 tribal dominated villages of 4 districts, namely Balangir, Dhenkanal, Kandhmal and Mayurbhanj. Table 1 shows that the total population of the villages was 6681. A total of 5750 (86%) eligible individuals of 12 villages in 6 districts clusters were included in the survey. In each cluster, male to female ratio was about 1 : 1.

### Prevalence of PTB based on screening of symptoms and sputum examination

Based on eligibility, 5144 (97.7%) individuals were screened for signs and symptoms of TB. About 139 (2.3%) could not be screened due to non-availability in their households during day time. Among those screened, 126 (2.4%) were found to have symptoms (Table 2).

Out of these, about 90% had cough of more than 2 weeks or in combination with other symptoms such as fever, chest pain, etc. and the rest 10% had other symptoms. Sputum samples were collected from 126 chest symptomatic individuals and sent to the National Reference Laboratory of this Institute, Bhubaneswar for further processing. As per the test reports, 35 patients were found to be having active TB disease. 18 were culture positive. Thus, the prevalence of pulmonary TB is 0.68%. All bacteriologically positive cases were referred for anti-TB treatment in their villages. Contacts of all those diagnosed with TB were also referred to the health authorities for TB investigations and follow up. Contacts of TB patients below 6 years were referred for TB prophylaxis as recommended by the TB programs.

**Table 1. Depicts the number of households & individuals surveyed, sputum samples collected from chest symptomatic persons & sputum/culture positive TB patients.**

| S. No. | District | Block | Name of Clusters | No. of Households surveyed | Individuals surveyed | Sputum from Chest Symptomatic persons | TB-ZN staining report | TB Culture positive |
|---|---|---|---|---|---|---|---|---|
| 1 | Mayurbhanj | Jharpokharia | Kasiabeda + Sankhabanka + Puranapani + Nuagoan | 288 | 889 | 49 | 12 | 09 |
| 2 | Mayurbhanj | Jharpokharia | Bhadua + Pandra | 234 | 819 | 26 | 10 | 04 |
| 3 | Mayurbhanj | Jashipur | Gandirabeda | 209 | 819 | 17 | 09 | 03 |
| 4 | Balangir | Patnagarh | Maghamara | 248 | 805 | 11 | 00 | 01 |
| 5 | Kandhamal | Tikabali | Penagaberi | 218 | 938 | 06 | 00 | 00 |
| 6 | Dhenkanal | Kamakhyanagar | Jantaribolo + Mahulapada + Mota | 203 | 874 | 17 | 01 | 01 |
| | Total | | | 1455 | 5144 | 126 | 32 | 18 |

**Table 2. Depicts the number of individuals surveyed, chest symptomatic persons & sputum / culture positive TB patients.**

| Total Individuals (n = 5144) | (n,%) |
|---|---|
| Symptomatic | 126 (2.44%) |
| TB Positive | 35 (27.77%) |
| Prevalence | 0.68% |

## Socio-demographic profile and cultural practices

The tribals of Mayurbhanj district were addicted to local made Handia and Mahuli (fermented rice water and other alcoholic beverages), bhang, khaini, etc. addiction is one of the reasons for non-adherence. Others being nausea and vomiting. Migration is an issue and the patients migrate to other neighbouring States for work. Sometimes the patient migrates without informing ASHA workers. The patients are neither aware of active TB case finding nor sputum examination for detection of TB. Sometimes, for contact screening, unwillingness of the family members of the patients are encountered by the Study team. Due to lack of awareness, people discontinue their medicines. STS were not visiting regularly the interior villages in some selected clusters. Lack of communication and poverty are the main factors due to which patients do not come personally to collect results. As people are poor, therefore, they prefer medicines, free of cost from government hospitals. Due to superstitious beliefs, the tribal people believe in black magic and some invisible divine powers. They do not take medicines and sacrifice animals with a belief to cure TB.

Malaria is prevalent in the villages of Mayurbhanj district. Each individual of every household had a history of Malaria. Malnutrition is common in Mayurbhanj. Fever, Malaria, Cough, Asthma, Arthritis and Diabetes are some of the common health problems of Jantaribola cluster in Dhenkanal and Kandhamal districts. Chronic Kidney Disease (CKD) is prevalent in Dhenkanal and Balangir districts. Skin diseases are more in Balangir as depicted in Table 3. A rare form of childhood obesity was seen among children in Jantaribola, Dhenkanal.

## Challenges faced by tribals of the selected clusters

As the selected clusters are present in the remote, dense and hilly forest regions of Odisha, there is no transport and communication facility to the nearest health facilities, the tribal people with any sort of illness prefer to linger their health issues till it gets serious. Basic facilities like safe drinking water and electricity are not available in some of the clusters. Some of the selected clusters are around more than 100 km from the district headquarters. There is lack of infrastructure and facilities at PHC levels. The tribal people in the selected villages were not aware about TB. They prefer to contact private practitioners, initially. If the health issue persists, then only they go to the nearby CHC, by walking or on their own arrangements.

**Table 3. Depicts some common diseases prevalent in the study sites.**

| Diseases | Districts | | | | | |
|---|---|---|---|---|---|---|
| | Balangir | Dhenkanal | Kandhamal | Mayurbhanj | | |
| | | | | Bhadua | Gandirabeda | Kasiabeda |
| Malaria | 28 | 74 | 65 | 121 | 108 | 412 |
| Diabetes | 13 | 18 | 24 | 6 | 2 | 4 |
| Hypertension | 29 | 61 | 53 | 2 | 3 | 19 |
| Arthritis | 3 | 66 | 82 | 22 | 25 | 05 |
| Asthma | 14 | 13 | 14 | 6 | 7 | 10 |

### Needs of the cluster

**For villagers of the selected study site.**   During the survey, we observed that awareness programs about TB, other infectious diseases, addiction to smoking, alcohol, its effect on health, hygiene, etc. are required for the villagers at regular intervals. Providing incentives to the tribal people coming from the remote and interior parts of the district may encourage them in accessing the health facilities. Arrangement of snacks and tea during FGDs and meetings with village heads, villagers encourages them to participate in interactive meetings.

**For chest symptomatics & TB patients.**   We have identified certain important issues which need to be explained and emphasized. The main among them are: i) screening and collection of sputum for testing,

ii) Contact screening of patients and their family members,

iii) treatment and the effects of discontinuing treatment and

iv) awareness programs about addiction, smoking, alcoholic beverages, etc. and its adverse effects on health.

**For health workers of the study site/district/state.**   Similarly, certain important issues which are needed for this group at regular intervals are:

i. Orientation of ASHAs, AWWs and other outreach workers for detection of chest symptomatic patients as ASHAs are the only health care workers easily accessible for the tribal community.

ii. Sputum collection camps to be organized where TB is more prevalent.

iii. Awareness programs on TB, Malaria and other non-communicable diseases, namely Diabetes, hypertension, etc.

## Discussion

In this study, 5144 (97.7%) individuals across 6 villages in the 4 tribal dominated districts of Odisha were screened for signs and symptoms of TB. Of these, 126 (2.4%) were having symptoms. 35 patients were found to be having active TB disease and 18 were culture positive. Thus, the prevalence of pulmonary TB is 0.68%. The risk factors seemed to be ignorance about TB symptoms, addiction to alcoholic drinks, difficulty reaching the health facilities owing to the long distances, lack of communication and transport. The family members of TB patients were neither willing for contact screening nor for Isoniazid prophylaxis to children. Ambulance facility is not available in the interior villages.

In addition, other morbidities like Malaria, diabetes, hypertension, malnutrition, CKD, etc. were observed in the tribes of the study sites. The functioning of RNTCP were reviewed and the gaps were identified. Several orientation training were organized for the health care providers, IEC activities were conducted in the villages for the villagers and the medical health workers.

There are two systematic reviews on prevalence of TB among tribes in India which suggest that more research needs to carried out for control of TB in tribal areas [11, 12]. Several authors have carried out TB prevalence studies in tribal communities of different regions of India as depicted in Table 4. Whereas some have focused on tribes of Madhya Pradesh namely, Saharia, Bharia, Baiga, etc., [13, 14, 15, 16, 17] others have done their studies in Car Nicobar, Kashmir valley, Tamil Nadu, Wardha of Maharashtra, West Bengal [18, 19, 20, 21, 22, 23, 24, 25, 26, 27, 28, 29]. Some others have focused on the yield of culture, symptoms questioning and sputum examination, chest X-ray and / or symptomatic screening [30, 31, 32].

**Table 4. Shows the comparative studies on prevalence of TB among tribals in different regions of India.**

| Sl. No. | Authors Name | Period and Place of Study | Study Population | Remarks |
|---|---|---|---|---|
| 1 | Balasubramanian et al [18] | Four sub-centres in Jamnamarathur Primary Health Centre area | Total population screened was 9383 persons; of these 5755 were aged 15 years and above | A total of 338 symptomatic subjects were identified; 12 sputum-positive cases were detected and started on treatment. |
| 2 | Basa et al [37] | This is a cross-sectional study, which had been carried out in Mayurbhanj district of Odisha during June-July, 2006 | 550 TB patients registered in 2005, covering all the seven TUs were included in the study, that is, 100% sampling. The case definition used was as per RNTCP guidelines: | Of the total 41 defaulters among 550 patients registered, only 31 could be interviewed, 10 were untraceable at the address provided. Default rate in our study was 7.5%. Majority of patients (73%) had defaulted during intensive phase of the treatment. A higher default rate associated with age group of 40–60 years, males and employed groups. The main reasons for default was due to drug toxicity (42%), feeling better so discontinued (35.5%), alcoholism (19.4%), migration (6.45%), wrong ideas (6.45%), DOTS provider absent (3.2%), DOTS provider rudeness (3.2%), and other reasons (9.7%), which includes family problems, timing inconvenient, and carelessness |
| 3 | Bhat et al [19] | The study was conducted amongst the tribal population of MP from July,2007—Feb.,2008 | Of the 23,411 individuals eligible for screening, 22,270 were screened for symptoms | The overall proportion of symptomatic individuals was 7.9%. Overall prevalence (culture and/or smear positive) of PTB was 387 per 1,00,000 population. The prevalence increased with age and was also significantly higher among males as compared with females. |
| 4 | Chakma et al [13] | Study was conducted in the primitive tribe of Sahariyas of Karhal block of Morena district, during the period Dec.,1991—June,1992. | The survey tor tuberculosis was done by the Regional Medical Research Centre (RMRC), Jabalpur among. 22,250 a total of 635 {5.7%} individuals had symptoms related to pulmonary tuberculosis. | Of these, 142 (22.4%) were sputum positive, thus, giving a relatively high overall disease rate of 12.7±2.1/1000. Infection rate among children aged below 9 years was also high, at 16.9±1.1%. Tribals were more prone to tuberculosis infection than non-tribals. Role of Various associated factors like frequent migration, socio-cultural behaviour, low humidity and dusty winds; of the area was considered and is discussed. |
| 5 | Chakrabarti et al [20] | A record-based cross-sectional study was conducted among patients registered in the TB register from Burdwan Medical College and Hospital from Jan.- Dec.,2009 in West Bengal. | The complete enumeration method was applied, and all TB patients were included in the study (n = 599). As the treatment outcome was available for all patients in the 2009 TB register, this was selected for the study. | Respectively 34.7% and 65.3% of the cases were tribals and non-tribals. Among tribal patients, 92.3% had pulmonary TB vs. 82.1% among non-tribals. The proportion of Category I cases (77.4%) was higher among tribals than among non-tribals (60.8%) |
| 6 | Chadha et al [21] | The study was carried out in the State Andhra Pradesh from 2005 and concluded in Mar.,2006 | A total of 3636 children, irrespective of their BCG scar status, were tuberculin tested using one TU PPD RT23 with Tween 80 and the maximum transverse diameter of induration was measured about 72 hours. The present survey was carried out among children between 5–9 years of age irrespective of their BCG status. | The prevalence of infection estimated by mirror-image technique using observed mode of reactions attributable to infection with tubercle bacilli at 20mm was 9.6%. The ARTI was computed at 1.4%. |

*(Continued)*

**Table 4.** (Continued)

| Sl. No. | Authors Name | Period and Place of Study | Study Population | Remarks |
|---|---|---|---|---|
| 7 | Datta et al [22] | Among 56 revenue divisions with a population of about 66 000, 24 revenue divisions were selected. The survey was conducted from May—Nov.,1989 Tiruvannamalai district, Tamil Nadu | There were 26,320 persons in the selected villages, of whom 7465 were aged under 10 years and were eligible for tuberculin testing, and 16,017 were over 15 years of age and were eligible for screening for disease. | Of the 6952 children tested and read, 5% had BCG scars and the prevalence of infection was 5%. The annual risk of infection was 1.1. Among adults, the prevalence of bacillary cases was 8/1000 and X-ray cases 29/1000. The prevalence of bacillary disease was higher among males, particularly with increasing age. Thirty symptomatic cases had normal X-rays and 63 X-ray cases had no symptoms. Thus prevalence would have been underestimated if either method had been used alone for screening. Isoniazid resistance was seen in 12% of patients, two of whom also had rifampicin resistance (2.6%). |
| 8 | Gopi et al [23] | A sample survey undertaken in Raichur district of Karnataka state | A population of 72,448 persons was registered. Of the 42,580 persons aged 15 years and above eligible for symptomatic and eligible for sputum examination. Sputum was collected from 3,685 (95.8%) of the 3,846 symptomatic and subjected to bacteriological examination | The number of symtomatics increased with increase in age, more often among males (11.9%) than among females (7.1%). The prevalence of tuberculosis, as assessed by smear and /or culture was 10.9 per 1,000 in population aged 15 years and above. The prevalence increased with age and was 3 times higher among males as compared to females. |
| 9 | Kolappan et al [40] | The study was carried out in Tiruvallur district In Tamil Nadu where DOTS. | Surveys of pulmonary tuberculosis were undertaken in representative samples of subjects aged >15 years (n = 83,000–92,000), initially and after two and half, five and seven and half years of implementation of DOTS | The prevalence of culture-positive tuberculosis was 607, 454, 309 and 388 per 100,000 in the four surveys, and that of smear-positive tuberculosis was 326,259, 168 and 180. In the first five years; annual decrease was 12.4% for culture-positive tuberculosis, and 12.2% for smear-positive tuberculosis |
| 10 | Mayurnath et al [25] | The Kashmir valley consists of three districts: Srinagar, Baramulla and Anantnag. The survey was conducted from June–Nov.,1978 | A tuberculosis prevalence survey was conducted in about 18,000 persons. Persons aged 5 yr and above were X-rayed (70 mm X-ray), and from such persons whose photofluorograms were interpreted as abnormal two specimens of sputum were collected and bacteriologically examined. In addition, a large X-ray of the chest was taken for children aged 0–4 yr who had reactions of 10 mm or more to PPD-S. | The results of the survey showed that the prevalence of non-specific sensitivity (59%) in the Kashmir valley is significant. The prevalence of tuberculous infection was 38 per cent. The prevalence of culture positive tuberculous patients (3 per 1000) and that of abacillary X-ray positive patients (14 per 1000) were found to be similar in the two sexes contrary to the usual experience of a higher prevalence among males. |
| 11 | Murhekar et al [26] | The study was carried out in 2001–2002 Andamans & Nicobars | Among the 4543 children enumerated, 4351 were tuberculin tested and read. | 981 children without bacilli Calmette-Guérin scars, 161 (16.4%) were infected with TB. A total of 77 cases who were smear-positive for TB were detected from among 10 570 people aged ≥15 years; the observed smear-positive case prevalence was 728.5 per 100 000. The standardized prevalence of TB infection, annual risk of TB infection, and prevalence of cases smear-positive for TB were 17.0%, 2.5%, and 735.3 per 100 000, respectively. |

(*Continued*)

**Table 4.** (Continued)

| Sl. No. | Authors Name | Period and Place of Study | Study Population | Remarks |
|---|---|---|---|---|
| 12 | Narang et al [27] | A study was undertaken from Sep., 1989—Nov.,1990 to the Ashti and Karanja tahsils in Maharashtra. | Prevalence study of pulmonary tuberculosis by house-to-house survey of symptoms among tribal ($n = 20{,}596$) and non-tribal ($n = 93{,}670$) populations aged 5 years and over. | The prevalence of smear and/or culture positive tuberculosis/100 000 population was 133 in the tribal and 144 in the non-tribal population. The difference in prevalence of symptomatic individuals and sputum positive cases among the tribal and the non-tribal populations was statistically significant only in the symptomatic individuals/100 000. The prevalence of cases in both groups was higher in males than females; however this difference was significant only in the tribal group. |
| 13 | Raj et al [29] | Field based TB surveys between 2004–2009 Madhya Pradesh | In total, 10,963 sputum smears were screened from Hindu tribes (n = 4032), Hindu non-tribal (n = 5445) and Muslim communities (n = 1486). | The prevalence of TB was found to be significantly higher in Hindu tribes compared with Hindu castes and Muslims. The overall RR of developing smear-positive disease was 1.4-fold higher in males than females in all the study groups. The highest prevalence of TB was observed in subjects aged 15–34 years. |
| 14 | Rao et al [15] | Study was carried out in the Baiga population in Baiga chak Dindori district in Chhattisgarh, during Jan.- Mar.,2008 | Villages in the area were selected randomly in order to cover the sample size of 2,100 with the study carried out in five villages. A population of 2,359 was covered under the study | Overall prevalence of PTB was 146 per 100,000 population |
| 15 | Rao et al [16] | A community-based cross-sectional TB prevalence survey was undertaken in the Saharia, a primitive tribal community of Madhya Pradesh. The study was carried out in the Karhal block, Sheopur District from Nov.,2007-Mar., 2008 | Of the 11,468 individuals eligible for screening, 11,116 (96.9%) were screened for symptoms. | The overall prevalence of pulmonary TB disease was 1518 per 100,000 populations. Prevalence increased with age and the trend was statistically significant. The prevalence of pulmonary TB was also significantly higher in males than females. |
| 16 | Rao et al [28] | A community based cross-sectional survey was undertaken in Jabalpur District of the central Indian state of Madhya Pradesh. This cross sectional study was conducted in the urban and rural populations of Jabalpur district from Jan.,2009—Jan.,2010 | Of the 99,918 individuals eligible for screening, 95,071 (95.1%) individuals were screened. | 7916 (8.3%) were found to have symptoms and sputum was collected from 7533 (95.2%) individuals. Overall prevalence of bacteriologically positive PTB was found to be 255.3 per 100,000 populations. Prevalence was significantly higher amongst males compared with females. Prevalence was also significantly higher in rural areas as compared to the urban. |
| 17 | Present study, Hussain et al | This study was conducted in tribal dominated districts of Balangir, Dhenkanal, Kandhamal and Mayurbhanj in Odisha during 2015–2017 | 5144 (97.7%) individuals were screened for signs and symptoms of TB. | 126 (2.4%) were found to have symptoms. 35 patients were found to be having active TB disease. 18 were culture positive. Thus, the prevalence of pulmonary TB is 0.68%. |

There are few studies on the various tribes of Odisha. These studies have focused on the socio-economic and cultural practices as well as ethno-medicines used by Bondo tribes of Malkangiri [33, 34]. There are hardly any epidemiological studies on prevalence of TB among the tribes in Odisha. Parija et al have studied the impact of awareness drives and community based active TB case finding in Odisha. Das et al have reported the factors leading to delay in diagnosis among pulmonary TB patients of Rayagada District. Basa et al have reported the number of default and its factors associated among TB patients treated under DOTS in Mayurbhanj district. Pati et al have determined the prevalence of Diabetes mellitus among newly diagnosed TB patients in Malkangiri district. Kamineni et al suggested measures to enhance advocacy, communication and social mobilization for TB control in Odisha [35, 36,

37, 38, 39]. Kolappan et al reported that the implementation of DOTS was followed by a substantial decrease in the prevalence of pulmonary TB in a rural community in south India over a period of seven and half years. They suggested vigilant supervision for sustaining the effectiveness of DOTS program. Subramani et al reported decline in TB prevalence by 50% in 5 years, from 609 to 311 per 100 000 population for culture-positive TB and from 326 to 169/100 000 for smear-positive TB. The annual rate of decline was 12.6% in south India [40, 41].

Studies from different countries namely China, Indonesia and Phillipines have reported that there is a decreasing trend in the prevalence of TB among the tribes. In China, it was 32%, during 1990–2000, in Philippines, it was 30% during 1997–2007 and in Indonesia, 67% during 1980–2004 and this variation may be due to the quality of screening [42, 43, 44].

India has up-scaled the basic services for TB in the health system where about 19 million TB patients under RNTCP were treated but the rate of decline of TB is too slow to meet the 2030 Sustainable Development Goals (SDG) and 2035 End TB targets. Previous efforts have resulted in inadequate declines and this is not likely to advance the progress towards eliminating TB. Novel comprehensive interventions are needed to increase the rate of decline of incident TB several fold, about 10–15% annually. The integration of four strategic pillars of Detect—Treat—Prevent—Build (DTPB) are needed for moving towards TB elimination [45].

Tuberculosis is a major problem among the tribal people though the prevalence is low. Strengthening of the program is required to reduce the prevalence of disease, early detection of all incident TB patients, multi-drug resistant patients, to ensure universal access to diagnosis and to achieve higher rates of treatment success.

## Strengths & limitations

The selected villages in the districts had less prevalence of TB. Other nearby villages had more number of Chest symptomatics and TB patients. Therefore, Medical Officers, STS and other health workers suggested to carry out the survey in other villages where the problem is more.

## Conclusion

The prevalence of TB is less among the tribal people in the selected districts of Odisha. But owing to the less awareness and difficulty in accessing health facilities, further strengthening of TB control programs are required. Repeat surveys, if carried out, in all the surveyed sites after an interval of approximately 5 years would provide information on future trends in different tribal dominated districts of the State. We, therefore, feel that screening of chest symptomatics for TB in tribal regions of the state, irrespective of their complaints, would go a long way in early detection of the TB.

## Supporting information

**S1 File. TB tribal forms.**
(RAR)

## Acknowledgments

We thank ICMR for funding this project grant (Tribal/89/TB-15/2014-ECD-II). We acknowledge the support of State TB Cell, Directorate of Health Services, Odisha at Bhubaneswar for granting us permission to conduct the study in the four districts of Odisha. We thank the State and District TB Officers, Dr.Sanjukta Sahoo, Dr. Kasturi Mishra, Dr. Bidyut Nanda, Dr. Jatin Pattanaik, Dr. Subhashis Mohanty and Dr. Bijay Kumar Das, Medical Officer of CHCs in Mayurbhanj district. We also thank Priyabrata Tripathy, STS and Sachin Barik, MPHS and

ASHAs, Aanganwadi workers for their assistance in carrying out the survey. The services rendered by Minaketan Barik and Niranjan Sahoo, Technicians are hereby acknowledged. Thanks are also due to Prakash Behera and Anakar Nayak who accompanied the study team to the different villages of the selected districts.

We are indebted to all the villagers of the study sites who have co-operated throughout the study period.

## Author Contributions

**Data curation:** Tahziba Hussain, Sushri Shanta Tripathy, Shritam Das.

**Formal analysis:** Tahziba Hussain.

**Investigation:** Sushri Shanta Tripathy, Shritam Das, Prakasini Satapathy, Dasarathi Das.

**Methodology:** Tahziba Hussain, Shritam Das.

**Project administration:** Tahziba Hussain, Sanghamitra Pati.

**Supervision:** Tahziba Hussain, Beena Thomas.

**Validation:** Tahziba Hussain, Sushri Shanta Tripathy.

**Writing – original draft:** Tahziba Hussain.

**Writing – review & editing:** Tahziba Hussain.

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
