## [Decision Letter · Decision Letter 0]

10 Jan 2020

PONE-D-19-33812

Prevalence, risk factors and health seeking behaviour of pulmonary tuberculosis in four tribal dominated districts of Odisha

PLOS ONE

Dear Dr. Hussain,

Thank you for submitting your manuscript to PLOS ONE. After careful consideration, we feel that it has merit but does not fully meet PLOS ONE’s publication criteria as it currently stands. Therefore, we invite you to submit a revised version of the manuscript that addresses the points raised during the review process.

We would appreciate receiving your revised manuscript. To enhance the reproducibility of your results, we recommend that if applicable you deposit your laboratory protocols in protocols.io, where a protocol can be assigned its own identifier (DOI) such that it can be cited independently in the future. For instructions see: http://journals.plos.org/plosone/s/submission-guidelines#loc-laboratory-protocols

We look forward to receiving your revised manuscript.

Kind regards,

Frederick Quinn

Academic Editor

PLOS ONE

Journal Requirements:

4. Your ethics statement must appear in the Methods section of your manuscript. If your ethics statement is written in any section besides the Methods, please move it to the Methods section and delete it from any other section. Please also ensure that your ethics statement is included in your manuscript, as the ethics section of your online submission will not be published alongside your manuscript.

5. We note that Figure #1 in your submission contains map images which may be copyrighted. All PLOS content is published under the Creative Commons Attribution License (CC BY 4.0), which means that the manuscript, images, and Supporting Information files will be freely available online, and any third party is permitted to access, download, copy, distribute, and use these materials in any way, even commercially, with proper attribution. For these reasons, we cannot publish previously copyrighted maps or satellite images created using proprietary data, such as Google software (Google Maps, Street View, and Earth). For more information, see our copyright guidelines: http://journals.plos.org/plosone/s/licenses-and-copyright.

a.    You may seek permission from the original copyright holder of Figure #1 to publish the content specifically under the CC BY 4.0 license. 

Reviewers' comments:

Reviewer's Responses to Questions

**Comments to the Author**

1. Is the manuscript technically sound, and do the data support the conclusions?

Reviewer #1: Yes

Reviewer #2: Yes

2. Has the statistical analysis been performed appropriately and rigorously? 

Reviewer #1: Yes

Reviewer #2: Yes

3. Have the authors made all data underlying the findings in their manuscript fully available?

Reviewer #1: Yes

Reviewer #2: Yes

4. Is the manuscript presented in an intelligible fashion and written in standard English?

Reviewer #1: Yes

Reviewer #2: Yes

5. Review Comments to the Author

Reviewer #1: A pertinent study looking at the prevalence of TB in tribal population in India. The study aims to understand the incidence of TB in the pupulation, along with their practices and helps in educating the population and alo to design better policies towards eradication of the disease

Reviewer #2: The authors seek to identify the influential factors that mediate the TB diagnostics and treatment. Based on the population size and incidence rate, the authors focused on 4 tribes in India. Targeting the selected subjects, representing significant portion of the tribal population, the study possess robust sample size, as well as comprehensive data collection. This study represents another meaningful example to understand how social and economic behaviors impact the effectiveness of anti-TB efforts. The significance is well observed. This reviewer recommends the following minor changes:

1. There needs some clarification in the Abstract to help readers easily comprehend the points. Line 37, out of these (?). Line 38, these (?). Line 40, are these 18 culture-positive among those 35 TB-active? Line 41-44, I could not understand what these sentences mean? Line 45, the conclusion sentence dose not convey any insightful conclusion.

2. The Introduction section needs to be cleaned. It is better to get to the point directly. In the manuscript, however, a lengthy introduction of tribal structure in India lead the way, from line 53 through line 70. The logic chain from line 79-89 is not clear. Try to follow the general structure, for example, what is the aim, what is the expectation, what is the strategy and method.

6. PLOS authors have the option to publish the peer review history of their article (what does this mean?). If published, this will include your full peer review and any attached files.

Reviewer #1: No

Reviewer #2: No

---

## [Author Response · Author response to Decision Letter 0]

23 Jan 2020

22nd Jan.,2020

Response to Reviewers

Dear Sir,

I thank you for reviewing the manuscript and giving me a chance to revise the same. I have incorporated the changes in the manuscript as suggested and highlighted in blue.

Thanking you & with regards,

Sincerely

Tahziba Hussain

Dr.Tahziba Hussain 

1. Please ensure that your manuscript meets PLOS ONE's style requirements, including those for file naming. The PLOS ONE style templates can be found at http://www.plosone.org/attachments/PLOSOne_formatting_sample_main_body.pdf and http://www.plosone.org/attachments/PLOSOne_formatting_sample_title_authors_affiliations. pdf

Ans. I have checked the manuscript for PLOS ONE's style requirements and made changes accordingly.

2. Please include additional information regarding the survey or questionnaire used in the study and ensure that you have provided sufficient details that others could replicate the analyses. 

Ans. The details of the questionnaire have been included in both the local language, Odiya and English, as Supporting Information.

Ans. The title has been amended. Comparison with studies in other regions of India has been added in the title in the online submission.

 4. Your ethics statement must appear in the Methods section of your manuscript. If your ethics statement is written in any section besides the Methods, please move it to the Methods section and delete it from any other section. 

Ans. Ethics approval has been given in the Methodology section.

5. We note that Figure #1 in your submission contains map images which may be copyrighted. All PLOS content is published under the Creative Commons Attribution License (CC BY 4.0), which means that the manuscript, images, and Supporting Information files will be freely available online, and any third party is permitted to access, download, copy, distribute, and use these materials in any way, even commercially, with proper attribution. For these reasons, we cannot publish previously copyrighted maps or satellite images created using proprietary data, such as Google software (Google Maps, Street View, and Earth). For more information, see our copyright guidelines: http://journals.plos.org/plosone/s/licenses-and-copyright.

Ans. The figure 1 containing the map has been removed from the manuscript.

1. There needs some clarification in the Abstract to help readers easily comprehend the points. Line 37, out of these (?). Line 38, these (?). Line 40, are these 18 culture-positive among those 35 TB-active? Line 41-44, I could not understand what these sentences mean? Line 45, the conclusion sentence dose not convey any insightful conclusion.

Ans. Line 37 - 40, the sentences have been re-framed. In Line 41-44, the risk factors have been elaborated. Conclusion sentence has been changed. 

2. The Introduction section needs to be cleaned. It is better to get to the point directly. In the manuscript, however, a lengthy introduction of tribal structure in India lead the way, from line 53 through line 70. The logic chain from line 79-89 is not clear. Try to follow the general structure, for example, what is the aim, what is the expectation, what is the strategy and method.

Ans. Line 79 - 89, the sentences have been re-framed including the aim, expectation, strategy and method.

---

## [Decision Letter · Decision Letter 1]

7 Feb 2020

Prevalence, risk factors and health seeking behaviour of pulmonary tuberculosis in four tribal dominated districts of Odisha : comparison with studies in other regions of India.

PONE-D-19-33812R1

Dear Dr. Hussain,

We are pleased to inform you that your manuscript has been judged scientifically suitable for publication and will be formally accepted for publication once it complies with all outstanding technical requirements.

With kind regards,

Frederick Quinn

Academic Editor

PLOS ONE

Additional Editor Comments (optional):

Reviewers' comments:

Reviewer's Responses to Questions

**Comments to the Author**

1. If the authors have adequately addressed your comments raised in a previous round of review and you feel that this manuscript is now acceptable for publication, you may indicate that here to bypass the “Comments to the Author” section, enter your conflict of interest statement in the “Confidential to Editor” section, and submit your "Accept" recommendation.

Reviewer #1: All comments have been addressed

Reviewer #2: All comments have been addressed

2. Is the manuscript technically sound, and do the data support the conclusions?

Reviewer #1: Yes

Reviewer #2: Yes

3. Has the statistical analysis been performed appropriately and rigorously? 

Reviewer #1: Yes

Reviewer #2: N/A

4. Have the authors made all data underlying the findings in their manuscript fully available?

Reviewer #1: Yes

Reviewer #2: Yes

5. Is the manuscript presented in an intelligible fashion and written in standard English?

Reviewer #1: Yes

Reviewer #2: Yes

6. Review Comments to the Author

Reviewer #1: comments have been addressed by the authors. Again this is a relevant work that addresses an important public health problem in developing countries

Reviewer #2: The authors have addressed all the comments from previous review. This reviewer does not have any further comments or concerns.

7. PLOS authors have the option to publish the peer review history of their article (what does this mean?). If published, this will include your full peer review and any attached files.

Reviewer #1: No

Reviewer #2: No

---

## [Editor Report · Acceptance letter]

27 Feb 2020

PONE-D-19-33812R1 

Prevalence, risk factors and health seeking behaviour of pulmonary tuberculosis in four tribal dominated districts of Odisha : comparison with studies in other regions of India. 

Dear Dr. HUSSAIN:

I am pleased to inform you that your manuscript has been deemed suitable for publication in PLOS ONE. Congratulations! Your manuscript is now with our production department. 

With kind regards,

on behalf of

Dr. Frederick Quinn 

Academic Editor

PLOS ONE